# Adverse Events to Comirnaty Vaccine Are Linked to Sex, Age and BMI: Should We Consider Reducing the Dose for Females?

**DOI:** 10.3390/vaccines11030505

**Published:** 2023-02-22

**Authors:** Elena Azzolini, Maximiliano Mollura, Chiara Pozzi, Leonardo Ubaldi, Alberto Mantovani, Carlo Selmi, Riccardo Barbieri, Maria Rescigno

**Affiliations:** 1IRCCS Humanitas Research Hospital, Via Manzoni 56, 20089 Rozzano, MI, Italy; 2Department of Biomedical Sciences, Humanitas University, Via Rita Levi Montalcini 4, 20072 Pieve Emanuele, MI, Italy; 3Politecnico di Milano, Department of Electronic, Information and Bioengineering, Piazza Leonardo da Vinci 32, 20133 Milano, MI, Italy; 4The William Harvey Research Institute, Queen Mary University of London, London EC1M 6BQ, UK

**Keywords:** vaccine, COVID-19, adverse events

## Abstract

An important issue that is often neglected is the difference between male and female genders in response to medical treatments. In the context of COVID-19 vaccine administration, despite identical protocol strategies, it has been observed that females often suffer more adverse consequences than males. Here, we analyzed the adverse events (AEs) of the Comirnaty vaccine in a population of 2385 healthcare workers as a function of age, sex, COVID-19 history and BMI. Using logistic regression analysis, we showed that these variables may contribute to the development of AEs, particularly in young subjects, females and individuals with a BMI below 25 kg/m^2^. Moreover, partial dependence plots indicate a 50% probability of developing a mild AE for a long period of time (≥7 days) or a severe AE of any duration in women below 40 years old and with a BMI < 20 kg/m^2^. As this effect is more evident after the second dose of the vaccine, we propose to reduce the amount of vaccine for any additional booster dose in relation to age, sex and BMI. This strategy might reduce adverse events without affecting vaccine efficacy.

## 1. Introduction

The impact of vaccines is influenced by both biological differences (sex) and social or cultural factors (gender) [1]. A growing number of studies have identified numerous immunological, genetic, hormonal and environmental factors that contribute to differences in vaccine responses and outcomes (such as efficacy and side-effects) based on sex and gender [2]. Cisgender females often show a more robust antibody response and, as a result, have higher efficacy and an increased likelihood of experiencing side-effects than males, highlighting the need for different dosing regimens based on sex [2,3]. For example, research on the influenza vaccine shows that women can achieve the same immunological response with a half-dose as men with a full dose [4]. Although sex clearly plays a role in COVID-19 disease and the vaccine response (including the immune response and likelihood of adverse events), few COVID-19 vaccine studies consider biological sex and perform sex-disaggregated analysis [5,6,7]. The SARS-CoV-2 mRNA vaccination (Comirnaty) has proven to be effective in protecting against hospitalization and death across the world [8,9]. The Comirnaty vaccine was administered in two doses (30 μg/dose) separated by at least 21 days in people 12 years of age and older. Although mRNA vaccines are particularly safe, they are not devoid of adverse events (AEs) [10,11,12,13]. In this study, we analyzed the incidence of AEs according to biological sex, age, COVID-19 history and body mass index (BMI) in response to the Comirnaty vaccination.

## 2. Materials and Methods

This clinical trial is a longitudinal study on 4227 healthcare workers (HCW) from 7 different healthcare facilities in Lombardy, Italy. The subjects were vaccinated with 2 doses of the Comirnaty vaccine between January and March 2021 and were asked to self-report any AEs on a questionnaire (see Appendix A). The reporting of adverse events followed the Italian and European reference legislation, in compliance with the Italian Strategic Plan for the anti-SARS-CoV-2/COVID-19 Vaccination and the Italian Medicines Agency (AIFA), which envisages the anonymous sharing of spontaneous reports through the IT platform of the National Pharmacovigilance Network provided by AIFA.

We considered all the variables of interest (sex, age, BMI and COVID-19 history) and the correct completion of the questionnaire. After excluding those patients with missing height and weight information, we analyzed 2385 subjects, of whom 13.8% had a COVID-19 history. We verified the validity of the analyzed population by evaluating the distribution of sex and COVID-19 history before and after removing patients with missing information. In particular, we observed that the incidence of female subjects was similar before (61.8%) and after the filtering procedure (60.8% for the first dose population and 62.2% for the second dose population). Similarly, the incidence of COVID-19 history was unchanged, equaling to 13.8% before and 12.4% (first dose population) and 13.9% (second dose population) after filtering, thus suggesting that we did not introduce any bias by the requirement of sex and BMI-related information.

The AEs after the first and second dose are reported in Table 1, *p*-values were calculated using the $\chi^{2}\;\mathrm{test}$, and the significance threshold was set to 0.001 according to Bonferroni’s correction to account for multiple AEs testing. We used the $\chi^{2}\;\mathrm{test}$ to evaluate the association between categorical variables, the Shapiro–Wilk test to check for normality, the ANOVA or Kruskal–Wallis test to evaluate differences in BMI when stratifying for sex and AEs, and a logistic regression analysis to evaluate the probability of AEs in relation to BMI while correcting for age, sex and COVID-19 history. Statistical significance was set for *p* < 0.05. Model improvement after the inclusion of BMI was assessed through the likelihood ratio test, and AEs probability varying age and BMI was analyzed by partial dependence plots for any symptoms equal or above the score of 4 for either males or females. We attributed a score of either 1 or 4 for a duration of 1 or 2 days, 2 or 8 for a duration of 3 to 6 days, and 4 or 16 for a duration ≥ 7 days for either mild or severe AEs, respectively. The analyses were conducted with Python 3.8.3.

## 3. Results

Table 1 reports the occurrence of mild or severe AEs in relation to sex, age, BMI and COVID-19 history after both doses of vaccine. Univariate testing of AEs after the first dose showed a significant association with local reactions at the injection site (*p* < 0.001). Local reactions at the injection site, asthenia, headache, muscle pain and joint pain AEs were significantly associated with the second dose (*p* < 0.001).

COVID-19 history was associated with an increased incidence of AEs only after the first dose (*p* < 0.001) and not after the second dose (*p* > 0.05). Importantly, the presence of AEs after the second dose was associated with AEs after the first dose (*p* < 0.001) in a matched subgroup receiving both administrations. When considering the whole population, BMI was associated with AEs incidence in both the first (*p* = 0.002) and second (*p* < 0.001) doses. In order to exclude sex effects in BMI, we separately tested male and female populations, showing that a lower BMI was linked to a higher incidence of AEs in males after the first dose (*p* = 0.0431), and in both males (*p* = 0.0482) and females (*p* = 0.0074) after the second dose (Figure 1A).

Table 2 shows the results of a logistic regression analysis, indicating that female and younger subjects are strongly associated with increased AEs after both vaccine doses (*p* < 0.001). For the first dose, there was no correlation between AEs and BMI. In contrast, BMI was inversely correlated with AEs after the second dose (*p* = 0.025), indicating that the second dose might be too high for young women with a low BMI. Additionally, having a history of COVID-19 is associated with increased AEs only for the first dose (*p* < 0.001). This implies that for individuals previously infected with SARS-CoV-2, the first dose may have a similar effect as the second dose for non-infected individuals.

BMI inclusion significantly improved model performance after the second dose (*p* = 0.023, likelihood ratio test). Interestingly, the AEs that mostly correlated with a lower BMI (<25 kg/m^2^) were local reactions (after both doses) and headache, muscle and joint pain (only after the second dose). We then evaluated, through partial dependence plots, the model’s output, i.e., the probability of developing an AE equal or above the score of 4 (meaning either a mild symptom lasting 7 or more days or a severe symptom for any duration of time), taking into account age, sex and BMI after the second dose. The partial dependence plots in Figure 1B show that one out of two females (below 40 years old and BMI below 20 kg/m^2^) are at risk of having long-lasting mild or any duration severe adverse events. Males are less susceptible to score 4 AEs.

## 4. Discussion

A recent study analyzed data from four cross-sectional studies on the Comirnaty vaccine’s AEs and found that females experience more AEs compared to males at all ages [13]. In our study, we explored the development of AEs while taking into account not only biological sex and age but also COVID-19 history and BMI variables in a logistic regression analysis. Our findings show that the Comirnaty vaccine induces AEs, particularly in young subjects, females, and individuals with a BMI below 25 kg/m^2^. Women under 40 years old and with a BMI < 20 kg/m^2^ had a 50% probability of developing a mild AE lasting for at least 7 days or a severe AE of any duration, especially after the second dose of the vaccine. These results suggest that biological sex, age and BMI differences should be considered when determining vaccine doses for booster shots, with a possible reduction in the dose for women under 40 years old and with a BMI < 20 kg/m^2^.

Fractional dosing has been successfully used for various diseases. For example, between 2016 and 2018, some countries utilized 1/5 doses of yellow fever vaccine to control epidemics following guidance from the World Health Organization (WHO) [14]. Immunogenicity data combined with model-based analysis for COVID-19 vaccines, developed by W. Wiecek and collaborators, suggest that half or quarter doses of some vaccines could be nearly as effective as current doses and more effective than other vaccines in use [15]. Moreover, fractional doses may induce fewer side effects, as suggested by clinical data on the ChAdOx1 nCoV-19 (AZD1222) vaccine [16]. For the BNT162b2 vaccine, 10 μg and 20 μg doses are predicted to have an efficacy of 70–85%, compared to roughly 95% for the standard dose [15]. These predictions need confirmation, and a randomized trial of immunogenicity comparing low dose (20 μg) with a full dose of BNT162b2 was recently completed (https://clinicaltrials.gov/ct2/show/NCT04852861, estimated study completion date: 30 September 2022). Lower doses of BNT162b2 vaccine have been tested and used for children between 5 and 11 years old, which should not affect vaccine efficacy but significantly reduce adverse events [17]. Even if fractional doses are less effective than standard doses, epidemiological analysis indicates that increasing vaccination in countries still facing supply constraints would decrease overall infections and deaths [15]. Reducing the dosage would save vaccine doses and make them available to many low- and middle-income countries where the vaccine is not yet accessible. In these settings, reduced doses of mRNA vaccines could potentially be more efficacious than the standard of care. From an ethical point of view, it is important to evaluate the best dose of the Comirnaty vaccine in relation to age, biological sex and BMI to obtain maximal protection while avoiding adverse events, without jeopardizing vaccine efficacy. 

## Figures and Tables

**Figure 1 vaccines-11-00505-f001:**
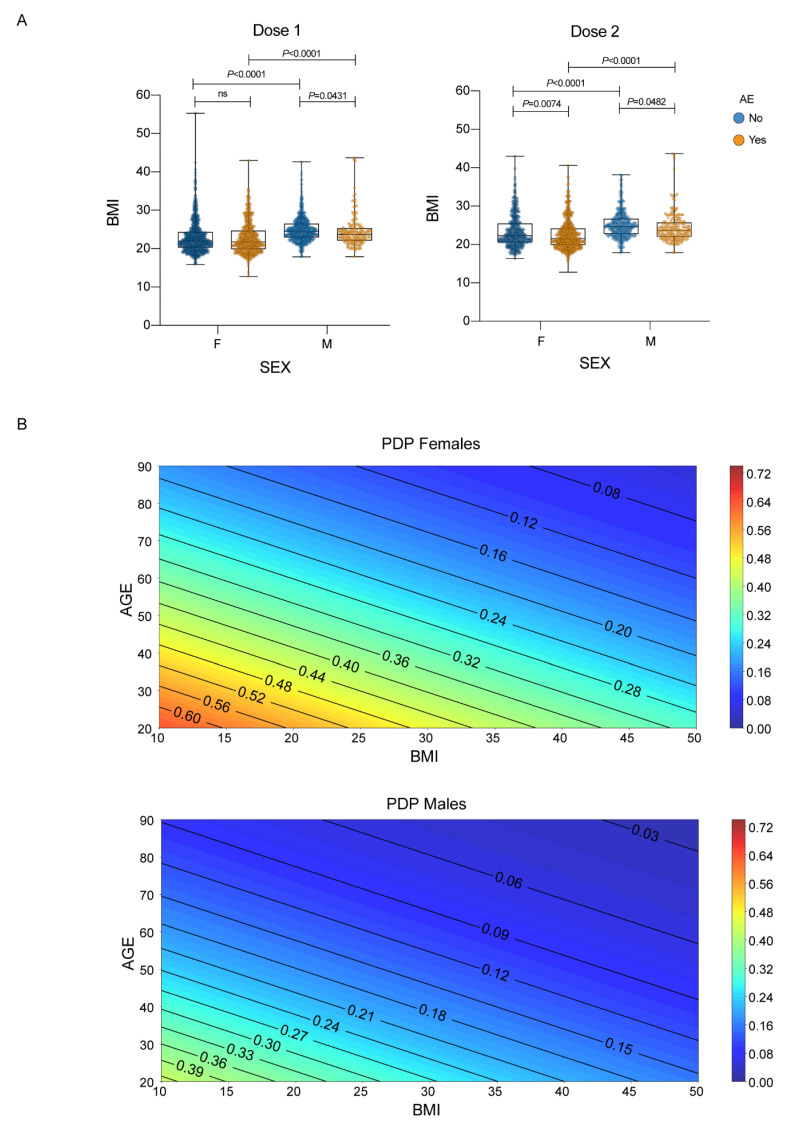
Sex, age, and BMI and their association with AE. (**A**) BMI distributions stratified by sex and any adverse events. Each dot corresponds to an individual subject. The box plots show the interquartile range, the horizontal lines show the median values, and the whiskers indicate the minimum-to-maximum range. *p*-values were calculated using Kruskal–Wallis test, as data were not normally distributed according to Shapiro–Wilk test, followed by Dunn’s test after the first and second dose; (**B**) Partial dependence plots of the logistic regression model (with age, sex and BMI as covariates) showing the probability of having either a mild AE for 7 or more days or a severe AE of any duration after the second dose of vaccine stratified by sex. Female subjects show higher risk at the same age and BMI.

**Table 1 vaccines-11-00505-t001:** AEs to Comirnaty vaccine reported after the first and second dose. Significant AEs are highlighted in bold. * means Severe AE, according to pharmacovigilance guidelines.

	DOSE 1					DOSE 2				
			BMI		χ^2^-Test			BMI		χ^2^-Test
	YES%	AE	≤25	>25	*p*-Value	YES%	AEs	≤25	>25	*p*-Value
Local reactions in the injection site	27.97	NO	985	444	**<0.001**	35.08	NO	686	356	**<0.001**
		YES	427	128			YES	435	128	
Asthenia	16.43	NO	1173	485	0.3855	33.71	NO	701	363	**<0.001**
		YES	239	87			YES	420	121	
Headache	12.20	NO	1228	514	0.0878	23.99	NO	822	398	**<0.001**
		YES	184	58			YES	299	86	
Muscle pain	8.47	NO	1286	530	0.2907	25.86	NO	804	386	**<0.001**
		YES	126	42			YES	317	98	
Joint pain	5.39	NO	1331	546	0.3400	20.12	NO	869	413	**<0.001**
		YES	81	26			YES	252	71	
Lymphadenopathy *	2.02	NO	1379	565	0.1551	7.35	NO	1036	451	0.6641
		YES	33	7			YES	85	33	
Chills	5.65	NO	1331	541	0.8652	19.38	NO	883	411	0.0053
		YES	81	31			YES	238	73	
Local Rash	1.01	NO	1403	561	0.0188	1.87	NO	1103	472	0.3245
		YES	9	11			YES	18	12	
Diffuse Rash *	0.20	NO	1408	572	0.4704	0.44	NO	1118	480	0.2516
		YES	4	0			YES	3	4	
Anxiety	0.30	NO	1406	572	0.2670	0.75	NO	1114	479	0.5779
		YES	6	0			YES	7	5	
Presyncope	0.96	NO	1396	569	0.3142	2.74	NO	1085	476	0.1122
		YES	16	3			YES	36	8	
Syncope *	0.00	NO	1412	572	1.0000	0.25	NO	1119	482	0.7486
		YES	0	0			YES	2	2	
Abdominal pain *	0.86	NO	1400	567	1.0000	3.24	NO	1084	469	0.9557
		YES	12	5			YES	37	15	
Insomnia	1.86	NO	1382	565	0.2459	5.42	NO	1058	460	0.6768
		YES	30	7			YES	63	24	
Diarrhea	1.36	NO	1394	563	0.7595	3.55	NO	1080	468	0.8396
		YES	18	9			YES	41	16	
Nausea	2.97	NO	1364	561	0.1079	7.73	NO	1022	459	0.0154
		YES	48	11			YES	99	25	
Vomiting	0.60	NO	1406	566	0.1922	1.25	NO	1103	482	0.0834
		YES	6	6			YES	18	2	
Angioedema (Facial swelling) *	0.10	NO	1411	571	1.0000	0.31	NO	1117	483	0.9939
		YES	1	1			YES	4	1	
Angioedema (Throat swelling) *	0.05	NO	1411	572	1.0000	0.25	NO	1119	482	0.7486
		YES	1	0			YES	2	2	
Transient facial paralysis *	0.20	NO	1408	572	0.4704	0.25	NO	1120	481	0.1582
		YES	4	0			YES	1	3	
Hypotension *	0.35	NO	1405	572	0.2044	0.75	NO	1112	481	0.9403
		YES	7	0			YES	9	3	
Sweating	1.97	NO	1381	564	0.3272	4.98	NO	1066	459	0.9253
		YES	31	8			YES	55	25	
Tachycardia *	1.01	NO	1397	567	0.8950	2.74	NO	1085	476	0.1122
		YES	15	5			YES	36	8	
Chest pain *	0.45	NO	1407	568	0.5043	1.43	NO	1101	481	0.1159
		YES	5	4			YES	20	3	
Dyspnea *	0.35	NO	1408	569	0.6871	1.18	NO	1108	478	1.0000
		YES	4	3			YES	13	6	
Fever (T < 38 °C)	2.82	NO	1367	561	0.1645	12.65	NO	964	438	0.0160
		YES	45	11			YES	157	46	
Fever (38 ≤ T < 39 °C)		NO	1405	567	0.5061	5.98	NO	1045	464	0.0526
	0.60	YES	7	5			YES	76	20	
Fever (T ≥ 39 °C) *	0.10	NO	1411	571	1.0000	1.93	NO	1098	476	0.7375
		YES	1	1			YES	23	8	

**Table 2 vaccines-11-00505-t002:** Results of logistic regression after first and second dose. Features showing significant association with AEs after the first or second dose are highlighted in bold.

AEs	Variable	Coefficient	*p*-Value
After First Dose	Intercept	−0.29	0.398
	Sex	0.82	**<0.001**
	BMI	−0.01	0.613
	Age	−0.02	**<0.001**
	COVID-19 History	0.85	**<0.001**
After Second Dose	Intercept	0.98	**0.007**
	Sex	0.75	**<0.001**
	BMI	−0.03	**0.025**
	Age	−0.02	**<0.001**
	COVID-19 History	−0.1	0.520

## Data Availability

According to the policy of the hosting Institution, data will be made available through Zenodo.

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
