# Peer review of "Adverse Events to Comirnaty Vaccine Are Linked to Sex, Age and BMI: Should We Consider Reducing the Dose for Females?"

_vaccines, 2023, doi:10.3390/vaccines11030505_

Round 1

Reviewer 1 Report

  1. The proposed manuscript “Adverse events to Comirnaty vaccine are linked to sex, age and  BMI should we consider reducing the dose for females?” is interesting paper.
  2. The introduction is too short and does not include the information about the vaccine doses. It would be interesting to know more about this parameter for other known vaccines and situation when the dose is corrected ( a few examples). Please include the information about basic principles of dosage ( if this information is available).
  3. The BMI abbreviation is not explained.
  4. The last section is not discussion, but conclusion. Please, change.

Author Response

REVIEWER 1

  1. The proposed manuscript “Adverse events to Comirnaty vaccine are linked to sex, age and  BMI should we consider reducing the dose for females?” is interesting paper.
  2. The introduction is too short and does not include the information about the vaccine doses. It would be interesting to know more about this parameter for other known vaccines and situation when the dose is corrected ( a few examples). Please include the information about basic principles of dosage ( if this information is available).

Thank you for your suggestion. We have extended the introduction by adding more information from previous studies that reviewed the different factors influencing vaccine response, especially in sex-based differences. Moreover, we have also included more information about the vaccine doses and the distance between first and second dose.

3. The BMI abbreviation is not explained.

Thank you for the suggestion. We have explained it.

4. The last section is not discussion, but conclusion. Please, change.

The reviewer is right. We have changed the last section.

Reviewer 2 Report

Thanks for submitting your brief manuscript. You are claiming that sex differences have significant impact on vaccine response. Can you show this using some other scholarly sources, not only one literature?

In the methods, kindly add information on the validity of the questionnaire. From 4277, to 2385, is it based on sex, age , BMI and COVID-19 history? If its the case, did you only take female participants? 

Why Kruskal Wallis test? Did you check the normality? Which statistical software was used?

You didn't mention anything related to ethical issues of the research?

Author Response

REVIEWER 2

Thanks for submitting your brief manuscript. You are claiming that sex differences have significant impact on vaccine response. Can you show this using some other scholarly sources, not only one literature?

Thank you for your suggestion. Now we have added additional references.

In the methods, kindly add information on the validity of the questionnaire. From 4277, to 2385, is it based on sex, age , BMI and COVID-19 history? If its the case, did you only take female participants? 

Thank you for your question. Most of the patients were dropped because of missing height and weight information. We verified the validity of the analyzed population by evaluating the distribution of sex and COVID-19 history before and after removing patients with missing information. In particular, we observed that the incidence of female subjects was similar before (61.8%) and after the filtering procedure (60.8% first dose and 62.2% second dose populations). Similarly, COVID-19 history incidence was unchanged equaling 13.8% before, and 12.4% (first dose population) and 13.9% (second dose population) after filtering, thus suggesting that we did not introduce any bias by the requirement of sex and BMI related information. We have also included this statement in the Methods section.

Why Kruskal Wallis test? Did you check the normality? Which statistical software was used?

Thanks for this important point. Yes, we have assessed normality of the data with the Shapiro-Wilk test, and after rejection of the null hypothesis that data come from a normal distribution, we have applied the Kruskal Wallis test. We have now updated the manuscript by reporting this information.

You didn't mention anything related to ethical issues of the research?

Thank you for your suggestion. We have mentioned this point in the discussion.

Reviewer 3 Report

The topic of gender differences in medicine is very important and it has extreme value.

Therefore, a work that analyzes the different AEs in females compared to males is good.

The paper reports the results of a higher prevalence (non-risk) of AE in women aged less than 40 years and with a BMI less than 20.

These results could correctly suggest setting up a "dose finding" study in this category of people to reduce AE but ensure the same efficacy.

Finally, the reduction of the dosage to make more vaccine doses available may not be correct, as here comes the issue of waste and vaccination solidarity  (in the West we have not had a lack of vaccine doses). 

In detail:

the initial sentence of the abstract (an important issue that is not usually considered is that male and females are different) is too simple. The Authors should value the Gender Medicine approach and say that in Medicine and in the response to medical treatments, the response between male and female gender is different.

Abstract. lines 20 and 21: "all these variables contribute to development of AE" means  that there is a causal relationship, while describing data on the prevalence of AE in some categories of people. 

Line 22: a "machine learning approach" is mentioned and it is not described in the "Methods" section. It would be interesting to know about this machine learning approach.

Lines 26 and 27. "this would drastically reduce adverse events without affecting vaccine efficacy". This is all to demonstare by means of results.

Introduction. line 31 "sex" should be substituted for "gender" in a Gender Medicine approach.

The introduction (according to the original article scheme given to this paper) should close with the objective of the study.

Materials and methods. line 39 it would be interesting to know if the 7 healthcare facilities have the same composition of healthcare personnel and therefore if they are comparable.

Line 41: "a questionnaire": it would be interesting to attach the questionnaire or at least explain how it is composed, if it has been validated in the literature or has envisaged a pilot validation phase during the study.

line 44: it could be interesting to specify the time interval between vaccination (1st and 2nd dose) and infection as this too could influence the onset of AE.

Line 46: what does 0.005/28 mean if immediately before it is said that the significance treshold has been set to 0.001?

Finally, the methods should correctly indicate that the reporting of adverse events (albeit collected with the questionnaire) followed the reference legislation according to the flows envisaged by the national pharmacovigilance system (Italian Medicines Agency - AIFA)

Results. lines 57-61: here there are methods described again.

Table 1: should be simplified (either only the YES values or only the NO and relative numerical values). Also, why enter the second dose first and then the first dose in the columns?

The p values of the first two lines are expressed in a different way than the following lines (e.g. 3E-06 and 2E-06). I think the method of expression of the numerical value of the p value should be standardized.

line 82: "BMI inversely correlates with AE only after the second dose". This aspect should be discussed in the discussion, considering that BMI probably does not change between the 1st and 2nd dose which are carried out 21 days apart.

In the discussion, the Authors state that the results may suggest a dose reduction as proposed in the work cited at n° 7 in the references. the cited study is a "dose finding" study in the pediatric population, while this study is an examination of the AEs and it is a gamble to state that the dose is at the basis of the different distribution of adverse events in the specific category of population (women <40 years with BMI <20) because other factors could also be involved.

Round 2

Reviewer 1 Report

The manuscript is improved and suitable for publication